# Association of a Bioimpedance Profile with Optical Coherence Tomography Features in Diabetic Macular Edema

**DOI:** 10.3390/jcm12206676

**Published:** 2023-10-22

**Authors:** Sunjin Hwang, Mincheol Seong, Min Ho Kang, Zheng Xian Thng, Heeyoon Cho, Yong Un Shin

**Affiliations:** 1Department of Ophthalmology, Hanyang University Guri Hospital, Hanyang University College of Medicine, Seoul 04763, Republic of Korea; sunjin1989@hanyang.ac.kr (S.H.); goddns76@hanmail.net (M.S.); bsdoc@hanyang.ac.kr (M.H.K.); 2Byers Eye Institute, Stanford University, Palo Alto, CA 94305, USA; thng_zheng_xian@ttsh.com.sg; 3National Healthcare Group Eye Institute, Tan Tock Seng Hospital, Singapore 308433, Singapore

**Keywords:** diabetic macular edema, hyper-reflective foci, InBody, optical coherence tomography, phase angle

## Abstract

We examined the association between bioimpedance profiles and optical coherence tomography (OCT) features in patients with diabetic macular edema (DME). This cross-sectional study included 100 eyes of 100 patients with type 2 diabetes mellitus. The systemic fluid status was assessed using extracellular water-to-total body water ratio (ECW/TBW) and phase angle (PhA), which was measured using bioimpedance equipment. ECW/TBW was higher in the DR (diabetic retinopathy) with DME group than in the no DR and DR without DME groups (*p* = 0.007 and *p* = 0.047, respectively); however, no significant difference was observed between the no DR and DR without DME groups. The PhA values were significantly lower in the DR with DME group (5.45 ± 0.84) than in the no DR (6.69 ± 0.69) and DR without DME groups (6.05 ± 1.15) (*p* < 0.001, *p* = 0.032, respectively). The presence of multiple HRF (hyper-reflective foci) was associated with a significantly higher ECW/TBW (*p* = 0.001). In the group with the most significant HRF, PhA was lower than in those with none or moderate amounts of HRF (*p* < 0.05). Bioimpedance fluid profiles of patients with OCT features of DME suggest a connection between the overall systemic state, including fluid status and DME development. Further research is required to fully understand and utilize this information for effective clinical assessment and treatment planning.

## 1. Introduction

The incidence of diabetes mellitus (DM) has surged globally, and its prevalence is expected to rise to 7.7% worldwide by 2030 [1]. Diabetic macular edema (DME) can adversely affect a patient’s vision; if untreated, 20% to 30% of patients with DME lose at least three lines of vision within 3 years [2]. The pathogenesis of DME is multifactorial; primarily, there is a disruption of the blood–retinal barrier (BRB), which causes fluid accumulation in the inner or outer layers of the macula [3]. Additionally, inflammatory processes, such as an increased vascular endothelial growth factor (VEGF) and protein kinase C levels, together with vascular endothelial dysfunction, can also cause DME; however, the exact details of DME pathogenesis remain to be fully elucidated.

Hyper-reflective foci (HRF) were first described as extravasated lipoproteins or proteins from the broken blood–retinal barriers in DME [4]. However, recent studies have suggested that HRFs can represent aggregations of activated microglial cells in DME and are associated with increased inflammation [5,6]. Bolz et al. found a positive correlation between the aqueous humor inflammatory biomarker soluble CD14 released by the retinal microglia and HRF [5]. The latter also serves as a valuable biomarker for the diagnosis and prognosis of patients with age-related macular degeneration, retinal vein occlusion, central serous chorioretinopathy, Stargardt’s disease, and retinitis pigmentosa [7,8]. However, there are no studies on the associations between HRF and systemic health.

The systemic body fluid status can be indirectly measured using a body composition monitor (BCM) and bioelectrical impedance analysis (BIA) [9]. As previously reported, impaired kidney function, characterized by albuminuria and increased creatinine, can predispose to DME [10]. Additionally, the optical coherence tomography (OCT) features of DME, such as central subfield retinal thickness (CSRT) and central subfield choroidal thickness (CSCT), can fluctuate under certain conditions, such as hemodialysis, which may alter the body fluid status abruptly [11]. The BIA can indirectly measure the body fluid status in a non-invasive manner. A study using the BIA in ophthalmology revealed that overhydration or volume overload is strongly associated with DME and suggested that the whole-body fluid status may alter the fluid status in the retina [12]. The phase angle (PhA) derived from BIA is a new indicator of general health status, muscle strength, and hydration status [13]. Previous studies have evaluated the utility of PhA in DM [14,15]; however, only a few studies have been conducted to determine the variations in PhA under ophthalmologic conditions [13,16]. In this study, we aimed to determine the association between the bioimpedance profiles and OCT features of DME.

## 2. Materials and Methods

### 2.1. Study Design

This was a cross-sectional observational study. All patients with a known history of DM who visited an ophthalmology clinic between January 2021 and January 2022 were enrolled. Written informed consent was obtained from all participants. This study was conducted in accordance with the Declaration of Helsinki and was approved by the Research Ethics Committee of Hanyang University Guri Hospital, Korea (Institutional Review Board No: 2021-03-014).

### 2.2. Participant Characteristics

Patients aged more than 18 years who were diagnosed with DM and could undergo OCT and BCM scans were recruited. The exclusion criteria included high refraction errors (>+6 diopters or <−6 diopters), intraocular inflammation, retinal diseases other than DM retinopathy (such as age-related macular degeneration, retinal vascular occlusion, macular hole, and epiretinal membrane), trauma, glaucoma, severe media opacity (poor image quality scores lower than 50), and a previous history of vitrectomy. All DR patients were recruited in a naïve state before receiving laser photocoagulation or intravitreal injections, and patients who had already received laser treatment or injections were excluded. Patients with severe systemic diseases, such as myocardial infarction, cerebral infarction or hemorrhage, type of cancer, and poor kidney function (estimated glomerular filtration rate (eGFR) < 60 mL/min/1.73 m^2^) were excluded. All patients underwent basic blood tests, including HbA1c, eGFR, and creatinine levels.

### 2.3. Ophthalmic Examinations

All patients underwent a swept-source OCT (SS-OCT) (DRI OCT-1 Triton^®^, Topcon Corporation, Tokyo, Japan) at a 1050 nm wavelength, a speed of 100,000 A-scans per second, an axial resolution of 8 µm, and a transverse resolution of 20 µm. The patients underwent a 12 mm × 9 mm three-dimensional wide scan with 256 B-scans, and each scan contained 512 A-scans (512 A-scans for each of the 256 B-scans). The SS-OCT images were viewed and analyzed using an Image Viewer (IMAGEnet 6, version 1.28.17646, Topcon Corp., Tokyo, Japan). The CSRT and CSCT were automatically measured, and the center of the fovea was automatically placed. In the case of segmentation errors, modifications were manually performed by two examiners (S.H. and Y.U.S) independently, and the average of the two values was used. The quantity of HRF was also counted by the two examiners on three horizontal OCT scans; one B-scan passed through the fovea, and two B-scans were individually 500 µm superior and inferior to the fovea. Only the HRFs located within 1 mm from the fovea center were counted, and hard exudates were not included (Figure 1). The HRF quantity was divided into three groups (absent 0; moderate 0–20; and significant ≥ 21) [17]. Intraclass correlation coefficients (ICCs) were calculated, and the agreement between the measurements was assessed. 

### 2.4. Body Composition Measurement

Every patient underwent a bioimpedance examination (InBody S10^®^, InBody Co., Ltd., Seoul, Republic of Korea) using a multi-frequency bioelectrical impedance analyzer at six different frequencies (1, 5, 50, 250, 500 kHz, and 1 MHz). The fluid status of the different body compartments was measured using an eight-electrode system connected to the fingertips and ankles. Various current frequencies from the different cell membranes pass through with different impedances and are translated into extracellular water (ECW) and intracellular water (ICW), providing information on total body water (TBW) [16]. The PhA was calculated using the resistance and reactance at 50 kHz. Using built-in software, we obtained data, including the ECW/TBW ratio of the whole body, percentage of body fat (PBF), body mass index (BMI), and PhA. A professional examiner calibrated the bioimpedance device daily according to the manufacturer’s recommendations. The patients underwent a bioimpedance examination for 5 min in the supine position after fasting for 8 h, refraining from extreme exercise for at least 24 h, and refraining from consuming alcohol or caffeine-containing beverages for at least 48 h immediately after the OCT examination.

### 2.5. Statistical Analysis

Patients were categorized into three groups: no diabetic retinopathy (DR), DR without DME, and DR with DME groups. Patients with DR or DME underwent a pre-specified subgroup analysis based on the amount of HRF present (absent, moderate, or significant, as defined previously) [18]. The average CSRT, CSCT, and HRF values for each patient were analyzed. Data with a normal distribution were expressed as mean ± standard deviation. An analysis of variance (ANOVA) was performed to compare BCM measurement variables and OCT features. Using the Shapiro–Wilk test, we could assume normality for all groups that did not include more than 30 patients. Therefore, we conducted a one-way ANOVA to perform the statistical analysis. Pearson’s correlation analysis was performed to determine the relationships among ECW/TBW, CSRT, and CSCT. All statistical analyses were performed using SPSS software (ver. 22.0 (IBM Corp., Armonk, NY, USA)). Statistical significance was set at *p* < 0.05.

## 3. Results

### 3.1. Baseline Characteristics

One hundred patients with DM were recruited for this study. The mean age of patients was 61.68 ± 11.02 years, and 48 of them (48%) were male. Details of the baseline characteristics, ophthalmic parameters, and bioimpedance parameters of the no DR (n = 24), DR without DME (n = 38), and DR with DME (n = 38) groups are summarized in Table 1. The repeatability of all measurements was sufficiently high (ICC > 0.9) for each examiner. Additionally, the reproducibility between the two examiners was high (ICC > 0.9).

### 3.2. Systemic Factors, Bioimpedance Parameters, and OCT Features in the No DR, DR without DME, and DR with DME Groups

Baseline characteristics of age, BMI, HbA1c, eGFR, and creatinine levels were well balanced across the three groups (*p* > 0.05). CSRT was thicker in the DME group (350.13 ± 79.16 µm) than in the other two groups (238.17 ± 24.02 µm, *p* < 0.001, 233.74 ± 24.99 µm, *p* = 0.001, respectively). CSCT was thicker in the DR without the DME group than in the No DR group (218.00 ± 43.61 µm vs. 258.29 ± 41.69 µm, *p* = 0.002), and the DR with DME group exhibited thicker CSCT than the other two groups (296.57 ± 37.79 µm, *p* = 0.001, and *p* < 0.001, respectively). The ECW/TBW value was higher in the DR with DME group than in the No DR and DR without DME groups (*p* = 0.007 and *p* = 0.047, respectively); however, there was no difference between the No DR and DR without DME groups (*p* = 0.790). As for PhA, the DR with DME group (5.45 ± 0.84) exhibited significantly lower PhA than that of the No DR group. Representative cases are shown in Figure 2.

### 3.3. Comparison of Systemic Parameters and Bioimpedance Parameters in the HRF Subdivision Groups (Table 2)

A subgroup analysis was performed for the DR group based on the number of HRF, including 29, 23, and 24 patients with absent, moderate, and significant HRF, respectively. The baseline characteristics were well balanced. Age, BMI, HbA1c level, eGFR, and creatinine levels were not significantly different among the three groups (*p* > 0.05). The ECW/TBW value was significantly higher in the significant-HRF group than in the absent-HRF group (*p* = 0.001). In addition, the PhA was lower in the significant-HRF group than in the absent-HRF and moderate-HRF groups (*p* < 0.05); however, the PhA did not differ between the absent-HRF and moderate-HRF groups (*p* = 0.391). In terms of macular edema, CSRT demonstrated no difference between the moderate HRF and significant HRF groups (*p* > 0.05); however, when compared to the absent-HRF group, both groups exhibited thicker CSRT (*p* < 0.05).

### 3.4. Correlation of ECW/TBW and CSRT, CSCT (Figure 3)

ECW/TBW was positively correlated with CSRT (R = 0.277, *p* = 0.009). In addition, CSCT was positively correlated with ECW/TBW (R = 0.301, *p* = 0.005).

## 4. Discussion

In the present study, we observed that body fluid volume overload, represented by ECW/TBW, was associated with macular edema and choroidal thickening in patients with DR and DME. The PhA levels were lower in the DR and DME groups than in the No DR group. In addition, we observed that HRF in patients with DR was correlated with PhA, which could indirectly reveal the general health status using BIA monitoring. The correlation between BRB breakdown and the development of macular edema has been explored in various studies [18]; however, whether the disruption of the BRB solely contributes to fluid volume expansion or whether a more complex bilateral interaction plays a role remains unclear.

As previously reported [19], ECW/TBW serves as a valuable biomarker of body fluid volume overload. Using BIA, Tsai et al. [20] observed that body fluid volume overload, represented as overhydration (a type of bioimpedance parameter), was closely related to DME, and they categorized DME into intraretinal cysts, subretinal fluid, and hard exudate to determine the difference in body fluid volume overload among them. Yao et al. [12] also used the same device, as used in our study, to measure body fluid volume overload with ECW/TBW and determined its relevance to DME, regardless of kidney function. A noteworthy finding of our study was the strong association between DME and the fluid overload measured using BIA. This significant correlation was observed even in cases where kidney function was similar among the groups. This emphasizes the potential importance of evaluating fluid distribution and volume regulation using BIA in the context of DME development. The observed association between fluid overload and DME highlights the possibility that imbalances in body fluid levels could contribute to the pathogenesis of DME independently of kidney function. These findings shed light on a novel aspect of the relationship between fluid overload and DME. They may open new avenues of research for better understanding and management of DME.

A noteworthy association between CSCT and ECW/TBW was observed, indicating the presence of body fluid volume overload. CSCT was observed to be significantly thicker in eyes with DME, consistent with a previous study [21] that excluded patients receiving local treatments, such as anti-VEGF injections or pan-retinal photocoagulation, which are known to induce choroidal thinning [22]. The exact pathophysiology underlying choroidal thickening in DME has not yet been elucidated. However, VEGF, a key cytokine that regulates vessel hyperpermeability, has been implicated in the development of DME, with some studies reporting higher levels of the VEGF protein in patients with DME [23]. This suggests a potential correlation between DME and increased VEGF levels, leading to augmented choroidal blood flow and subsequent CSCT thickening. Our findings offer insights into the relationship between CSCT, body fluid volume overload represented by ECW/TBW, and DME, although the sequence of events regarding whether DME precedes CSCT thickening or vice versa remains unknown. Nevertheless, these results enhanced our understanding of the mechanisms underlying DME with respect to bodily fluid overload.

The DR without DME and DR with DME groups demonstrated lower PhA than that of the No DR group. The PhA has been analyzed as a marker of cell quantity, and the integrity of the cell membrane and lower PhA may explain the reduced viability of individual cells. Some studies have reported that the PhA can be used as an alternative method to measure an individual’s nutritional and overall health status and can be utilized as a useful tool for predicting clinical illness [24]. The DR group exhibited a lower PhA than the No DR group. Notably, in animal models of DR, apoptosis of non-vascular cells has been identified in the earlier stages [25]. In a longitudinal study, the researchers observed retinal degeneration, such as thinning of the retinal nerve fibers and ganglion cell layers (retinal neurodegeneration), in the absence of vascular pathology [26]. Additionally, a previous study [27] highlighted the role of oxidative stress in DR pathophysiology. As PhA reflects cell viability and is known to reveal the extent of oxidative stress, a lower PhA in the DR group seemed plausible in our study. Several studies have reported differences in the PhA between patients with DM and controls. Nsamba et al. observed that children with type 1 DM exhibited a lower PhA than that of age-matched controls (4.94 vs. 5.32, *p* < 0.001) [28]. Dittmar et al. [29] reported that the whole body PhAs of patients with diabetes at 50 and 100 kHz were lower than those of controls; however, the PhA at 5 kHz was higher in patients with diabetes. Furthermore, Kim et al. [15] observed that patients with diabetes exhibited a lower PhA at three frequencies (5, 50, and 250 kHz) than that of age-, sex-, and BMI-matched normal controls. However, all these studies were limited by their small sample sizes.

Furthermore, the DR with the DME group demonstrated a lower PhA than that of the DR without the DME group. One of the pathogenesis mechanisms of DME is the breakdown of the BRB. The BRB regulates the passage of proteins, ions, and water to sustain a perfect retinal microenvironment and protect the retinal function as a neuron. Following the breakdown of the BRB, the balance between hydrostatic and oncotic pressures is disrupted, and macular edema progresses [30]. The BRB loses its protective function owing to the impediment of cells and cell-to-cell junctions that comprise the BRB, namely pericyte loss, impaired cell-to-cell junctions, and basement membrane thickening of the capillary [31]. These events appear to reduce cell viability and may explain the low PhA observed in patients with DME.

We subdivided the DR group into the following three subgroups based on the number of HRFs: absent, moderate, and significant. There was a trend towards a lower PhA as the number of HRFs increased. Recent studies on HRF have reported that HRF serves as an aggregate of microglial cells that are strongly associated with inflammation [6,32,33]. Lee et al. observed that CD14 pro-inflammatory cytokine, expressed by microglia, macrophages, and monocytes, is strongly correlated with HRF [5]. Another study using a non-obese diabetic mouse model demonstrated that pro-inflammatory cytokines induce HRF and upregulate microglial cells [34]. Tomeleri et al. [35] also observed that the PhA can act as a predictor of inflammatory and oxidative stress, regardless of age and body composition. The group with significant HRF exhibited lower PhA in instances where a heightened active inflammation was present; alternatively, this event could be attributed to the compromised BRB, which consequently led to its destruction, indicating lower cell viability and contributing to lower PhA, as indicated in a previous study [36]. Reduced PhA levels are also observed with volume overload or anemia in heart failure diseases [37] and in underweight patients, such as those with anorexia nervosa. The PhA is a good predictor of impaired prognosis (mortality, post-operative complications, longer hospital stay, disease activity, and progression) of diseases, such as pancreatic, breast, and lung cancer; HIV infection; systemic sclerosis; bacterial sepsis; liver cirrhosis; and surgery [38,39,40]. As many studies have reported PhA as an effective biomarker of many diseases, it would be worth exploring the relationship between PhA and HRF in future studies.

This study had some limitations. First, the sample size was relatively small, which may have resulted in inadequate statistical analysis. Second, we recruited participants who were relatively healthy and whose eGFR was > 60 mL/min/1.73 m^2^, which may have caused a selection bias. However, the study’s strengths lie in its cross-sectional design and immediate OCT examination facilitated by a skilled examiner. Third, our bioimpedance profile measurements, particularly the PhA measurement, revealed only slight differences among the groups. Although the differences in the PhA between the groups were statistically significant, they may not be considered clinically significant because the variations within the normal range were small. Nevertheless, this study was the first to compare HRF with PhA, and it can serve as a pilot study for future investigations. Further studies with larger cohorts and standardized devices are warranted to validate and confirm these findings. Fourth, we were unable to assess changes in HRF and OCT findings in relation to BIA measurements following improved DME treatment. This study lacked a longitudinal follow-up of patients to track the evolution of both BIA and OCT parameters over time. Additional research and longitudinal studies are needed to enhance the robustness of these associations.

Nevertheless, to our knowledge, this study is the first to compare HRF with PhA and can serve as a pilot study for future investigations. Moreover, another strength lay in its cross-sectional design and immediate OCT examination facilitated by a skilled examiner. Further studies with larger cohorts and standardized devices are required to validate these findings.

## 5. Conclusions

Fluid volume overload, identified using BIA, is linked to macular edema and choroidal thickening in patients with DR. Lower PhA, potentially indicating health and inflammation status, was observed in patients with DR, DME, and abundant HRF, correlating with a more severe DR condition. Bioimpedance is a convenient, non-invasive, and economical method for assessing body composition and fluid volume. Our study combined bioimpedance measurements with OCT findings, highlighting the interplay between ocular health and overall systemic conditions. These findings underscore the importance of systemic health improvements in the treatment of DR and DME.

## Figures and Tables

**Figure 1 jcm-12-06676-f001:**
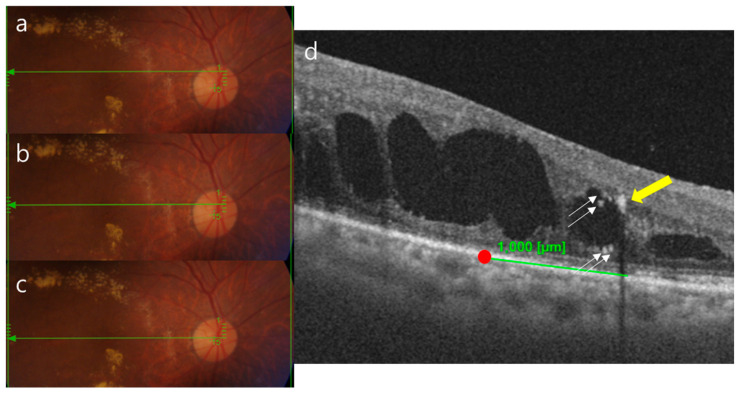
Schematic diagram related to counting the number of HRF. Acquisition of OCT slabs at the fovea center (**b**), 500 μm above the fovea center (**a**), and 500 μm below the fovea center (**c**). The red dot represents the fovea center, and only the HRFs located within 1 mm from the fovea center were counted (**d**). The white arrow indicates HRFs, while the yellow arrow points to hard exudates identified via posterior shadowing; these exudates were not included in the count. HRF, hyper-reflective foci.

**Figure 2 jcm-12-06676-f002:**
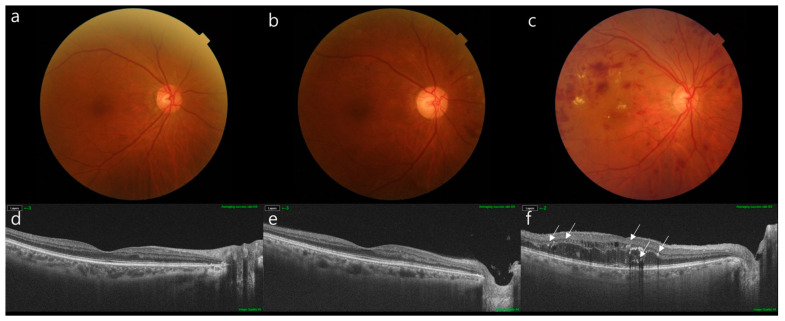
Color fundus photographs and OCT B-scan images of representative cases in each group. Color fundus photograph (**a**) and OCT B-scan image (**d**) of a patient without DR. The patient’s ECW/TBW was 35.71%, and PhA was 10.3. The CSRT and CSCT were 193 µm and 250 µm, respectively. The white arrow indicates the location of the HRF on the OCT image. Color fundus photograph (**b**) and OCT B-scan image (**e**) of a patient with severe NPDR but no DME. The ECW/TBW and PhA of the patients were 38.11% and 6.2, respectively. The CSRT and CSCT were 217 µm and 260 µm, respectively. Color fundus photography (**c**) and OCT B-scan images (**f**) of patients with PDR and DME. The ECW/TBW and PhA were 40.74% and 5, respectively. The CSRT and CSCT were 383 µm and 290 µm, respectively. CSRT, central subfield retinal thickness; CSCT, central subfield choroidal thickness; DR, diabetic retinopathy; DME, diabetic macular edema; ECW/TBW, extracellular water-to-total body water ratio; NPDR, non-proliferative diabetic retinopathy; PDR, proliferative diabetic retinopathy; and PhA, phase angle.

**Figure 3 jcm-12-06676-f003:**
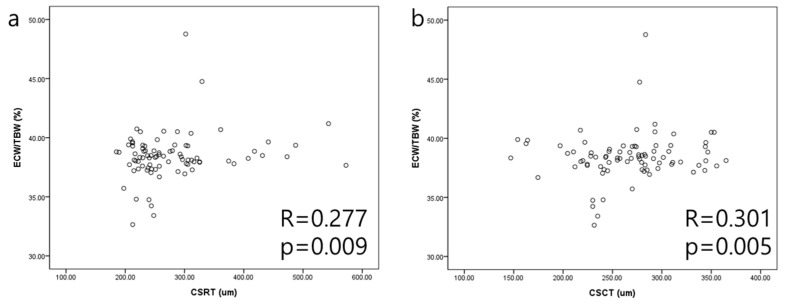
Scatter plots representing the relationship between ECW/TBW and retinal and choroidal thickness. Scatter plot depicting the correlation between ECW/TBW and CSRT (**a**). A positive correlation was observed between ECW/TBW and CSRT (R = 0.277, *p* = 0.009). ECW/TBW also reveals a positive correlation with CSCT (R = 0.301, *p* = 0.005). (**b**) CSRT, central subfield retinal thickness; CSCT, central subfield choroidal thickness; ECW/TBW, extracellular water-to-total body water ratio.

**Table 1 jcm-12-06676-t001:** Demographic and clinical characteristics of patients.

Characteristics	No DR, n = 24	DR without DME, n = 38	DR with DME, n = 38	*p* ^a^	*p* ^b^	*p* ^c^
Age, years	59.26 ± 8.51	57.06 ± 11.09	60.75 ± 9.94	0.204	0.309	0.325
BMI, kg/m^2^	23.20 ± 2.91	24.39 ± 3.33	23.68 ± 2.12	0.282	0.577	0.806
HbA1c, %	7.99 ± 0.94	8.25 ± 2.09	8.27 ± 1.76	0.101	0.508	0.304
eGFR, mL/min/1.73 m^2^	82.40 ± 15.08	82.75 ± 15.54	77.87 ± 14.63	0.998	0.680	0.754
Creatinine, µmol/L	0.86 ± 0.22	1.11 ± 1.05	1.06 ± 0.56	0.435	0.967	0.563
**Ocular parameters**						
CSRT, µm	238.17 ± 24.02	233.74 ± 24.99	350.13 ± 79.16	0.950	0.001	<0.001
CSCT, µm	218.00 ± 43.61	258.29 ± 41.69	296.57 ± 37.79	0.002	0.001	<0.001
**BCM parameters**						
ICW (L)	20.98 ± 3.79	25.31 ± 5.46	22.42 ± 4.28	0.003	0.039	0.491
ECW (L)	12.79 ± 2.01	15.15 ± 3.09	14.48 ± 2.45	0.004	0.558	0.05
TBW (L)	33.77 ± 5.66	40.46 ± 8.32	36.90 ± 6.55	0.003	0.114	0.236
ECW/TBW, %	37.99 ± 1.77	37.58 ± 2.31	39.35 ± 2.49	0.790	0.007	0.047
PhA (°)	6.69 ± 0.69	6.05 ± 1.15	5.45 ± 0.84	0.048	0.032	<0.001

BCM, body composition measurement; BMI, body mass index; CSRT, central subfield retinal thickness; CSCT, central subfield choroidal thickness; DR, diabetic retinopathy; DME, diabetic macular edema; eGFR, estimated glomerular filtration rate; ECW, extracellular water; HbA1c, glycosylated hemoglobin; ICW, intracellular water; PhA, phase angle; TBW, total body water. *p*
^a^: *p*-value between No DR group and DR without DME group; *p*
^b^: *p*-value between DR without DME group and DR with DME group; *p*
^c^: *p*-value between No DR group and DR with DME group.

**Table 2 jcm-12-06676-t002:** Comparison among the DR groups divided by HRF quantity.

Characteristics	Absent HRF, n = 29	Moderate HRF, n = 23	Significant HRF, n = 24	*p* ^a^	*p* ^b^	*p* ^c^
Age, years	57.58 ± 10.86	61.12 ± 10.45	59.26 ± 10.52	0.993	0.269	0.197
BMI, kg/m^2^	24.44 ± 3.44	23.94 ± 1.92	23.46 ± 2.19	0.870	0.530	0.269
HbA1c, %	8.17 ± 1.92	7.71 ± 1.79	8.00 ± 2.12	0.581	0.363	0.821
eGFR, mL/min/1.73 m^2^	83.65 ± 23.05	81.88 ± 26.21	73.64 ± 28.47	0.973	0.611	0.386
Creatinine, µmol/L	1.08 ± 1.07	0.96 ± 0.48	1.19 ± 0.65	0.894	0.710	0.901
ICW	25.55 ± 5.21	21.60 ± 4.26	23.03 ± 4.73	0.270	0.306	0.015
ECW	15.25 ± 3.01	13.71 ± 2.37	15.04 ± 2.62	0.812	0.778	0.439
TBW	40.81 ± 7.96	35.32 ± 6.53	38.07 ± 7.14	0.430	0.446	0.058
ECW/TBW, %	37.47 ± 2.31	38.94 ± 1.85	39.68 ± 2.86	0.146	0.084	0.001
PhA	6.13 ± 1.08	5.68 ± 0.94	5.20 ± 0.81	0.391	0.005	0.001

BMI, body mass index; DR, diabetic retinopathy; HRF, hyper-reflective foci; ECW, extracellular water; eGFR, estimated glomerular filtration rate; HbA1c, glycosylated hemoglobin; HRF, hyper-reflective foci; ICW, intracellular water; PhA, phase angle; TBW, total body water; *p*
^a^: *p*-value between no HRF group and moderate HRF group; *p*
^b^: *p*-value between moderate HRF group and many HRF group; *p*
^c^: *p*-value between no HRF group and many HRF group.

## Data Availability

The data that support the findings of this study are available from the corresponding author upon reasonable request.

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
