# Peer review of "Association of a Bioimpedance Profile with Optical Coherence Tomography Features in Diabetic Macular Edema"

_jcm, 2023, doi:10.3390/jcm12206676_

Round 1

Reviewer 1 Report

The authors present a study relating retinal and choroidal biomarkers on OCT slabs to biomedical impedance in patients with DM2. Some previous research has already been done, but the use of new technologies such as swept-source OCT is always interesting. Introduction, methods and discussion sections are fine, but there are just some issues to revise.

1) The study design is cross-sectional, but not prospective because no follow up has been performed. This appears both in abstract and methods section.

2) Sample size is lower than 30 in at least one group. Therefore, non-parametric test would be better used in this occasion: Kruskal Wallis, Spearman's correlations...

3) Figure 2 should be corrected. The scatter plots tend to a horizontal line. A few distant values are responsible for the present slope. That is why scatter plots are created, in order to assess outcomes visually.

Author Response

Response to Reviewer 1 Comments

Point 1: The study design is cross-sectional, but not prospective because no follow up has been performed. This appears both in abstract and methods section. 

Response 1: We would like to thank the reviewer for this comment. As you mentioned, this study has no follow up data of the patients. We changed the error in the manuscript and deleted it (both abstract and methods section).

Point 2: Sample size is lower than 30 in at least one group. Therefore, non-parametric test would be better used in this occasion: Kruskal Wallis, Spearman's correlations...

Response 2: Thanks for your comment. We did not specify in the manuscript that we performed a Shapiro-Wilk test for normality. In the Shapiro-Wilk test, it was assumed that the three groups are normally distributed, so a one-way ANOVA was used in our study. We added this content in the manuscript, too.

Point 3: Figure 2 should be corrected. The scatter plots tend to a horizontal line. A few distant values are responsible for the present slope. That is why scatter plots are created, in order to assess outcomes visually.

Response 3: The values of ECW/TBW are clustered between 37-40, so the scatter plot tends to look like a straight line. I have removed the outliers, recreated the chart, and attached it to the manuscript.

Reviewer 2 Report

Dear Authors , I found your Article very original and of great scientific significance. Correlation of diabetic macular edema and overall body fluid status together with general health is a new perspective that could lead to innovative approaches to this worldwide sight-treathening disease. 

I only suggest you some minor revisions, mainly text editing corrections : 

in the abstract few abbreviations are not previously elucidated , like 

line 15  - DR 

line 19 - HRF 

I also have a question regarding the exclusion criteria : 

- did you exclude pseudophakic patients or only patients with intraocular inflammation due to recent surgery? how did you assess intraocular inflammation? Did you use direct Anterior chamber inflammatory cells count (laser flare photometry ) or  Hyper- reflective Foci counts as an indirect measure? 

Author Response

Response to Reviewer 2 Comments

Point 1: in the abstract few abbreviations are not previously elucidated , like

line 15  - DR

line 19 – HRF 

Response 1: Thank you for your comment. We added full terms in the abstract which was not previously elucidated in the manuscript. DR : Diabetic Retinopathy; HRF : Hyperreflective foci.

 Point 2: I also have a question regarding the exclusion criteria :

- did you exclude pseudophakic patients or only patients with intraocular inflammation due to recent surgery? how did you assess intraocular inflammation? Did you use direct Anterior chamber inflammatory cells count (laser flare photometry ) or  Hyper- reflective Foci counts as an indirect measure?

Response 2: We did not exclude pseudophakic patients but did exclude those who had undergone vitrectomy. Among our patient group, there were no individuals who had undergone cataract surgery within the last year. We excluded cases of intraocular inflammation that occurred due to uveitis and similar conditions, rather than inflammation resulting from surgery. We did exclude cases where cells appeared in the anterior chamber, but as you mentioned, we did not measure it using laser flare photometry. We did not consider HRF as intraocular inflammation. Thanks again for the comment.

Reviewer 3 Report

Thanks for offering the opportunity to review this paper.  The authors tried to evaluate the association between bioimpedance profiles and OCT features in DME. However, there are certain issues with this paper that need to be addressed.

1. In the abstract part, the author directly gives the abbreviations of some terms without giving the full names.

2. Are there units for the BMC parameter in Table 1?

3. The DR patients included by the authors are NPDR or PDR?

4. Have these patients received any relevant treatment, such as laser treatment or anti-VEGF treatment, before they were included in the study?

5. In Part 2.3, the authors should give a schematic diagram to explain how they count the number of HRFs.

6. The conclusion part is too long.

Minor editing of English language required

Author Response

Response to Reviewer 3 Comments

Point 1: In the abstract part, the author directly gives the abbreviations of some terms without giving the full names.

Response 1: Thank you for the comment. We directly used the abbreviations in the abstract part, such as DR, and HRF. We added full terms in the abstract of DR and HRF. DR (Diabetic Retinopathy), HRF (Hyperreflective foci).

Point 2: Are there units for the BMC parameter in Table 1?

Response 2: Thank you for the good comment. It seems that adding the units for the BMC parameters has been omitted. The body water content like ICW, TBW, ECW, etc., can be expressed in liters (L), and the phase angle can be expressed in degrees (°) as it is an angular measurement. We added this unit in Table 1.

Point 3: The DR patients included by the authors are NPDR or PDR?

Response 3: That's an excellent point. When we initially designed our research, we ran statistics distinguishing between NPDR and PDR. There was a difference in body water content between No DR and DR, but no significant difference appeared between NPDR and PDR. In conclusion, DR patients consisted of nearly equal proportions of NPDR and PDR. Although it does not appear that changes in body water content vary according to the staging of DR, we believe that additional follow-up studies are needed to confirm this.

Point 4: Have these patients received any relevant treatment, such as laser treatment or anti-VEGF treatment, before they were included in the study?

Response 4: Thank you for the insightful comment. It appears that we inadvertently omitted this information from the exclusion criteria. The BCM test was conducted on both NPDR and PDR patients in their naive state, before any injections or laser treatments were administered. We added this contents in exclusion criteria.

Point 5: In Part 2.3, the authors should give a schematic diagram to explain how they count the number of HRFs.

Response 5: As you suggested, including a schematic diagram to illustrate the method of counting HRFs would be beneficial. I have added content regarding this, as Figure 1, in the method section. Thank you.

Figure 1. Schematic diagram related to counting the number of HRF. Acquisition of OCT slabs at the fovea center (b), 500 um above the fovea center (a), and 500 um below the fovea center (c). The red dot represents the fovea center, and only the HRFs located within 1mm from the fovea center were counted (d). The white arrow indicates HRFs, while the yellow arrow points to hard exudates, identified by posterior shadowing; these exudates were not in-cluded in the count.

Point 6: The conclusion part is too long.

Response 6: You’re correct, the conclusion section seems lengthy and could potentially lose its impact. I've attempted to revise it for conciseness. Thank you. 

Fluid volume overload, identified using BIA, is linked to macular edema and choroidal thickening in patients with DR. Lower PhA, potentially indicating health and inflamma-tion status, was observed in patients with DR, DME, and abundant HRF, correlating with a more severe DR condition. Bioimpedance is a convenient, non-invasive, and economical method for assessing body composition and fluid volume. Our study combined bioim-pedance measurements with OCT findings, highlighting the interplay between ocular health and overall systemic conditions. These findings underscore the importance of sys-temic health improvements in the treatment of DR and DME.

Reviewer 4 Report

The aim of the study titled " Association of bioimpedance profile with optical coherence tomography features in diabetic macular edema" was to evaluate the association between bioimpedance profiles and optical coherence tomography (OCT) features in diabetic macular edema (DME) based on results obtained from 100 patients with type 2 diabetes mellitus.

Diabetic retinopathy is a global health concern on the rise in developed countries, thus of important clinical interest. Macular lesions have become of increasing interest in ophthalmology in the past decade, especially in the midst of the constant evolution of OCT instrumentation and numerous treatment options with intravitreal injections and anti-VEG treatments.

The study is well-structured, with distinct headers and subheadings that help with material organization and linking particular issues. Each parameter is highlighted to draw attention to how important it is. The work is interesting and has clinical applications. The tables and figures are helpful in explaining the findings and are pertinent. Figures with clinical illustrations are instructive and detailed. The topic is current, and the references are appropriate. It can be of therapeutic relevance to briefly discuss available treatments and how these parameters change following treatment.

A native English doctor's editing can improve the text's English and flow.

Author Response

Thank you for your insightful comments and the meticulous attention to detail. I have completed the necessary corrections and enhancements as per your suggestions. Your expertise has significantly contributed to elevating the professional quality of my paper. I am truly grateful for your invaluable assistance.

Round 2

Reviewer 1 Report

The authors have made all suggested changes. However, it still appears in the text that the study was prospective instead of cross-sectional. It appears in page 9 line 313, and in page 9 line 328.

Author Response

Thank you again for your thorough comment. We've changed our manuscript based on your comment. 

However, the study's strengths lie in its prospective design and immediate OCT examination --> However, the study's strengths lie in its cross-sectional design and immediate OCT examination

Moreover, another strength lay in its prospective design and immediate OCT examination facilitated by a skilled examiner. --> Moreover, another strength lay in its cross-sectional design and immediate OCT examination facilitated by a skilled examiner. 

Reviewer 3 Report

Minor editing of English language required

Minor editing of English language required

Author Response

I have entrusted an English proofreading specialist company to have my document proofread by a native English speaker, and I will also provide a proofreading certificate along with it. However, if you find any awkward or unnatural parts while reviewing the proofread document, please leave comments so that I can make the necessary revisions. I appreciate your comments to enhance the readability and professionalism of the paper.
